# Feasibility of a silicone vascular phantom replicating real arterial contrast filling dynamics on cerebral angiography: An in-vitro pilot study

Ki Baek Lee[1,☯], Jae Jon Sheen[2,☯], Jong-Tae Yoon[3], Mi Hyeon Kim[3], Jun Young Maeng[3], Sun Moon Hwang[3], Joon Ho Choi[3], Deok Hee Lee[3]*

1 Department of Radiologic Technology, Chungbuk Health & Science University, Cheongju, Republic of Korea, 2 Department of Neurosurgery, Uijeongbu St. Mary's Hospital, College of Medicine, The Catholic University of Korea, Seoul, Republic of Korea, 3 Department of Radiology, Research Institute of Radiology, Asan Medical Center, University of Ulsan College of Medicine, Seoul, Republic of Korea

☯ These authors contributed equally to this work.
* dhlee@amc.seoul.kr

**Data Availability Statement:** All relevant data are within the manuscript

**Funding:** D.H.L; This work was supported by the Advanced Technology Center Plus (ATC+) Program

## Abstract

Some cerebral arterial silicone phantoms have been used in preclinical evaluations. However, typical silicone-based phantoms are limited in their capacity to reproduce real contrast filling dynamics of the human cerebral artery. This study aimed to develop a cerebral arterial silicone phantom to analyze the feasibility of real contrast filling dynamics. The fluid circulation phantom system consisted of a cerebral arterial silicone phantom without or with additional devices, a pump, an injection system, a pressure-monitoring system, a constant-temperature bath, and a venous drainage container. Vascular resistance was reproduced with a plastic cistern only or a plastic cistern filled with a sponge pad. Three phantom groups were constructed as follows: a) the cerebral arterial silicone phantom used as the control group (type A), b) phantom with the incorporated plastic cistern (type B), and c) phantom with the incorporated plastic cistern filled with a sponge pad (type C). The contrast concentration–time curve patterns of the three groups obtained from digital subtraction angiography (DSA) were compared. Consequently, the DSA pattern of the type C phantom was the most similar to that obtained from the control group as the reference data, which showed the broadest full-width-at-half-maximum and the area under the curve values and the highest maximum contrast concentration. In conclusion, we could emulate the arterial contrast filling dynamics of clinical cerebral angiography by applying a small cistern filled with a sponge pad at the drainage side of the phantom.

## Introduction

Understanding the concentration change of the contrast medium (CM) in cerebral angiography is vital in determining the relationship between contrast concentration and the timing of

(Grant numbers: 20009512) funded by the Ministry of Trade, Industry & Energy (MOTIE, Korea), and by the Technology Development Program (Grant numbers: S2836138) funded by the Ministry of SMEs and Startups (MSS, Korea) The funders had no role in study design, data collection and analysis, decision to publish, or preparation of the manuscript.

**Competing interests:** The authors have declared that no competing interests exist.

cerebral blood circulation. However, research on this subject is complicated, as it is ethically inappropriate to repeatedly inject CM into a patient and expose them to radiation. Ahmed et al. represented the bolus geometry of intra-arterial flow by injecting CM into animals [1]. A major limitation of that study was that the hemodynamic conditions among the experimental animals could not be kept constant. After injecting the CM into the animal, recirculation of the CM throughout the animal body may affect the results of subsequent experiment batches [2]. In addition, ethical considerations limit animal testing [3]. In this context, cerebral arterial silicone phantoms that can maintain stable conditions and control external confounding factors such as a constant heart rate, body temperature, and blood pressure would be useful for preclinical evaluation. Various *in vitro* silicone vascular phantoms have been successfully used in previous preclinical studies [4–8]. However, one of the limitations associated with silicone phantoms connected with a closed fluid circuit was the difficulty in emulating real contrast filling dynamics identified in cerebral angiography in real clinical settings. This is attributed to the fact that most of the silicone vascular phantoms were developed based on a large arteriovenous shunt without any peripheral resistance, irrespective of whether they were connected to peristaltic or pulsatile pumps.

Several studies have represented contrast concentration-time curves (CCTCs) with CM at the internal carotid artery (ICA) [9,10]. The characteristic of this CCTC graph was that the slope from the CM injection to the maximum intensity point was steeper than it from the maximum intensity point to washing-out. In other words, the CM was rapidly filled up to the maximum intensity point, but the pattern progressed slowly when the CM was drained out because of peripheral vascular resistance. We endeavored to emulate the peripheral vascular resistance of the precapillary arterioles of the brain parenchyma by applying an additional structure in between the silicone vascular phantom segment and the draining side of the closed circuit.

The objective of this study was to determine the feasibility of developing a cerebral arterial silicone phantom to replicate real contrast filling dynamics of conventional cerebral angiography.

## Materials and methods

### Patient selection

This study was approved by Asan Medical Center Institutional Review Board (No. 2020–1013). Informed consent was waived because of the retrospective nature of the study and the analysis used anonymous clinical data. We analyzed 376 patients who underwent intraarterial catheter angiography in our center from March 1 to June 30 in 2018. The exclusion criteria were as follows: the presence of intracranial stenosis, arteriovenous fistula, or arteriovenous malformation, which can affect the cerebral circulation time; and any other disease, except for an unruptured aneurysm. After the application of the exclusion criteria, 30 cases remained had the same graph pattern as the studies mentioned above [9,10]. Therefore, one patient data among 30 cases was used as a control to represent the shape of the entire CCTC graphs to compare the CCTC with a cerebral arterial silicone vascular phantom (Fig 1). The silicone vascular phantom was manufactured with three-dimensional (3D) printing and mold techniques using the 3D volume data of the selected representative patient. An additional component was attached to reproduce the wash-out of the actual arterial phase. The detailed fabrication description using 3D printing is presented in the "Cerebral arterial silicone phantoms composition" section.

An Artis zee biplane interventional radiology and fluoroscopy system (Siemens Healthineers) was used for intraarterial catheter angiography. A 4-Fr diagnostic angiocatheter was

376 patients who were performed with intraarterial catheter angiography

Exclusion criteria:
Intracranial stenosis, arteriovenous fistula, and arteriovenous malformation, which can affect the cerebral circulation time.

30 cases with an appropriate control group that can represent the shape of the entire CCTC graphs for comparing the CCTC with a cerebral arterial silicone vascular phantom

The absence of any other diseases, except for an unruptured aneurysm

One patient data with an MCA aneurysm was selected as a control group

**Fig 1. Flow diagram of patient selection.**

placed at the level of the C3–C4 vertebrae for the DSA imaging. A power injector (Medrad Mark V ProVis; Bayer healthcare) was used to inject 8 mL of CM (Visipaque, iodine 270 mgI/ mL, GE Healthcare) from a 50 mL bottle at a rate of 4 mL/s for two-dimensional (2D) angiography. This same CM injection protocol was used for experiments with the three types of cerebral arterial silicone phantoms (CASPs).

## Image processing with the iFlow

Quantitative comparisons of contrast concentration variations were performed as time progressed in the cerebral artery based on Syngo iFlow (Siemens Healthineers), an image processing software. The Syngo iFlow color-coding tool was used for 2D-DSA acquisitions. In these analyses, the color represented the transit time [11]. With these data, a CCTC of the CM bolus at selected locations within these 2D-DSA images could be created (Fig 2). A coronal plane of the cerebral DSA images for the Syngo iFlow was serially obtained at 7.5 frames/s over a period of 4 s, 4 frames/s for 6 s, and 2 frames/s. To avoid any vascular overlapping, we foundthe terminal ICA in the frontal plane was the most appropriate portion in measuring the contrast concentration.

## Cerebral arterial silicone phantoms composition

A CASP vessel model was created from 3D angiography of the control group data. After identification of the region-of-interest (ROI) on 3D angiography images, the vessel was segmented based on the intensity (Hounsfield units), and modeled with the use of the software Mimics 17.0 (Materialize NV), Meshmixer 2.0 (Autodesk), and Aview 2010. Segmented vessels were converted to a stereolithography file format for 3D printing (Fig 3A).

Subsequently, the CASP constructed based on the blood vessel model was developed by a 3D printer (Zortrax M200). The vessel model was printed with the copolymer acrylonitrile butadiene styrene at a resolution of 90–390 μm. The surface of the output was sprayed with

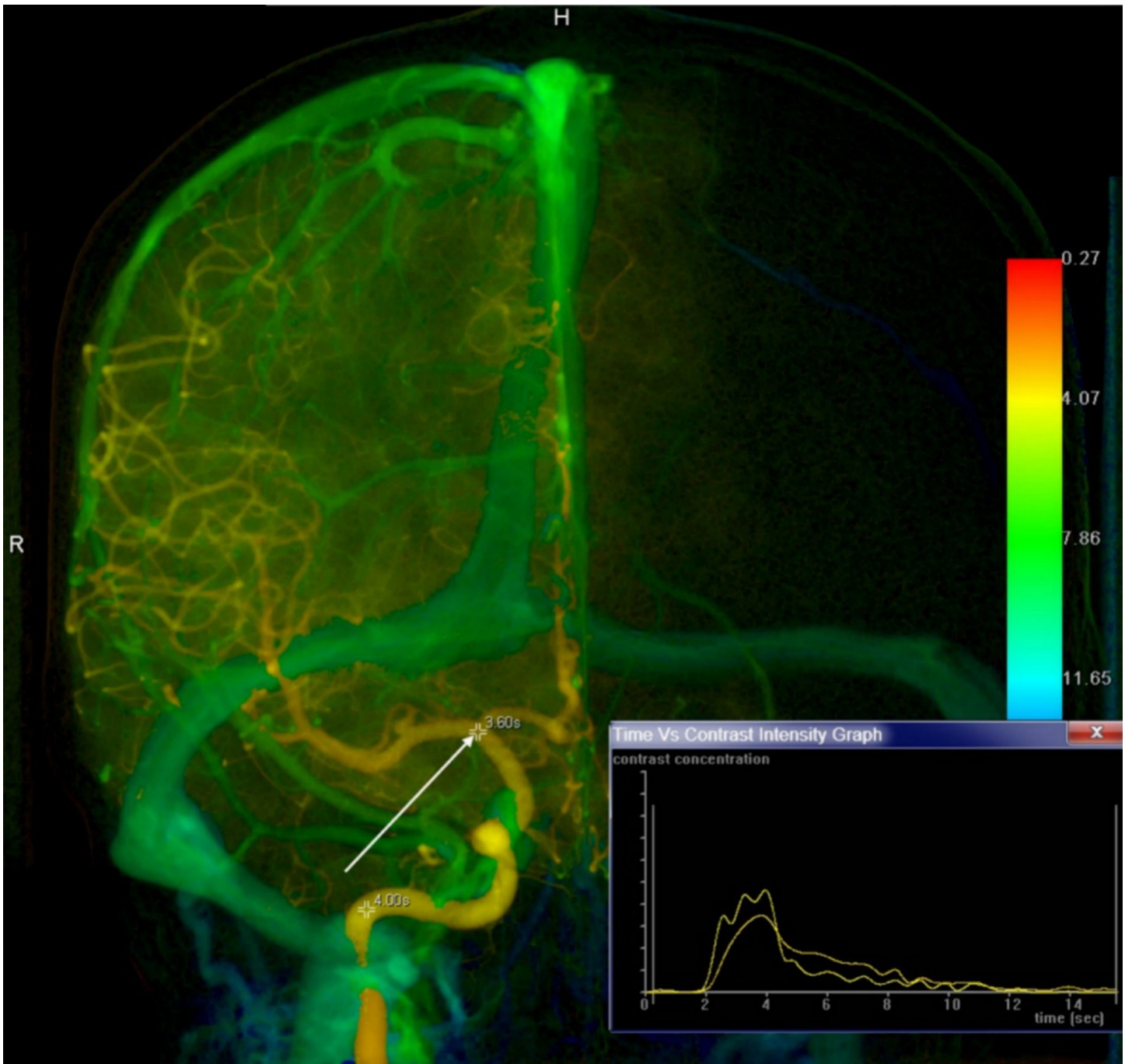

**Fig 2. Display of the region-of-interest in color-coded digital subtraction via the anteroposterior (AP) view.** The contrast concentration was measured at the terminal internal carotid artery (*white arrow*) in the AP view.

acetone. Additionally, sandpaper and varnish solution were used to trim the 3D-printed vascular model. The varnish solution was used to fill the irregular fine grooves of the vascular model surface. Subsequently, the vascular model was immersed in liquid silicone (Psycho paint; smooth-on, USA) (Fig 3B). After the silicone had solidified, it was cut off to remove the vascular model, leaving a hollow silicone mold (Fig 3C and 3D). Molten wax was injected into the

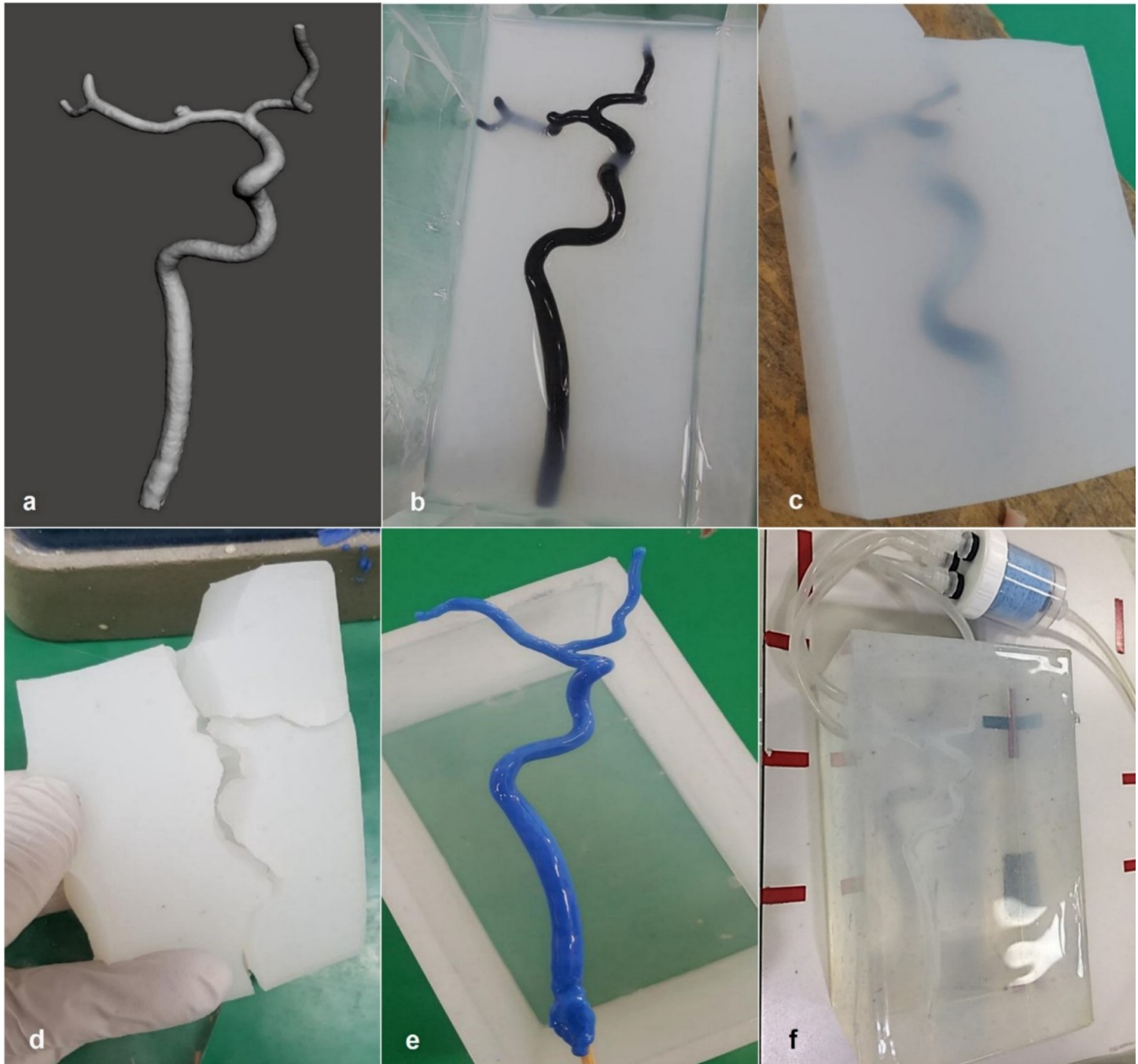

**Fig 3. Cerebral arterial silicone phantom (CASP) used to replicate real contrast filling dynamics on cerebral angiography.** (a) A three-dimensional (3D) vessel image in a stereolithography file format used for 3D printing. (b) A 3D-printed vascular acrylonitrile butadiene styrene copolymer model in a container containing liquid silicone. (c and d) Elimination of the surrounding silicone and extraction of the 3D-printed model. (e) The silicone-coated wax model was placed in rectangular silicone housing. (f) The resistance module of the fluid circulation CASP was composed of a plastic cistern and was filled with a sponge pad.

silicone mold to obtain a vascular wax model. Subsequently, a thin silicone layer was applied to the wax model approximately five times. The silicone-coated wax model was placed in a rectangular silicone housing (Fig 3E). Holes were punched in the silicone housing to allow a connection with the silicone-coated wax model. Transparent jelly-type silicone was then

poured into the silicone housing to complete the hexahedral cerebral arterial phantom, after which the mold was placed in an oven to melt the wax. Finally, a CASP was constructed in the shape of the control group's cerebral artery (Fig 3F).

## Configuration of the fluid circulation cerebral arterial silicone phantoms

Our fluid circulation CASP system consisted of a CASP with or without additional devices, a pump, an injection system, a pressure-monitoring system, a constant-temperature bath, and a venous drainage container. The entire platform is shown in Fig 4. To mimic blood circulation, water was transferred from a constant-temperature (37°C) bath to the venous drainage container by a peristaltic flow pump (Ecoline VC-380; ISMATEC). Additionally, the physiological parameters (pulse rate, blood pressure, viscosity, body temperature) of the actual patient were measured to reproduce an accurate hemodynamic environment, and were used as the input data for the fluid circulation CASP. The peristaltic pump was

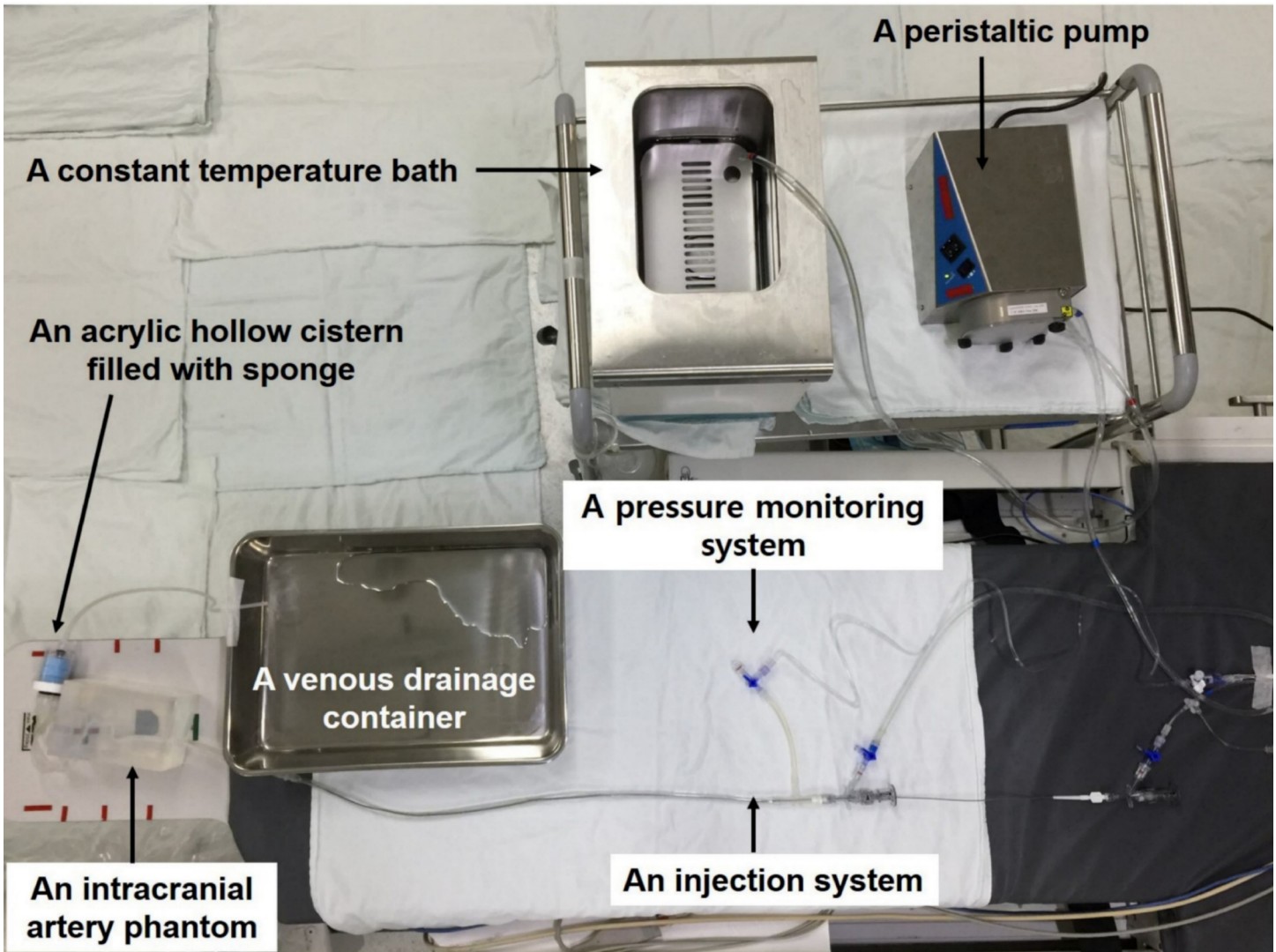

**Fig 4. Photograph of the entire fluid circulation cerebral arterial silicone phantom system.**

controlled to generate a pulse rate of 60 beats per minute (bpm) and a systolic pressure of 100 mmHg in the platform, similar to that of the patient. A 40:60 glycerol/water mixture for fluid viscosity was applied as the working fluid [12]. In fact, the dynamic viscosity of the blood-simulating fluid produced in the above-mentioned experiment was 3.22 [12], similar to the patient's dynamic viscosity. Therefore, we utilized the same dilution to obtain the same viscosity. The dynamic viscosities between the actual patient's blood and the fluid circulation CASP were $3.00 \times 10^{-3}$ kg/m·s and $3.22 \times 10^{-3}$ kg/m·s, respectively. A venous sinus pressure of approximately 10 cmH$_2$O was achieved by elevating the venous drainage container to a height of 10 cm above the overall system height. Iodine-based CM was injected via a 4-Fr catheter with a power injector, similar to the contrast injection method applied to the control group. Lastly, a plastic cistern filled with a sponge was utilized to represent real peripheral resistance. Both were cylindrical in shape, and the specifications were as follows: plastic cistern: the circle had a radius of 1.7 cm, the height of 4.2 cm, and a total volume of 38.11 cm$^3$; a sponge pad: porosity of 50 pixels per inch (ppi), a circle with a radius of 1.6 cm, and a height of 3.3 cm, with a total volume of 26.53 cm$^3$, and a density of 1.76 μg/mm$^3$ (Fig 3F). Real contrast filling dynamics were designed to control the resistance of the phantom's flow system as follows: a) the CASP was considered as the control group (type A), b) the CASP with the incorporated plastic cistern (type B), and c) CASP with the incorporated plastic cistern filled with a sponge pad (type C).

## Evaluation of the cerebral arterial silicone phantoms

To evaluate the fluid circulation in the CASP, CCTC patterns from the Syngo iFlow system were compared (control group vs. three phantom types). The intraarterial catheter-based angiograms were conducted using similar X-ray projection angles and catheter positions. The iFlow images were obtained at the same frame rate and projection angle.

Four parameters of the CCTC were adopted to analyze the iFlow images as follows: maximum contrast concentration, time-to-peak, full-width-at-half-maximum (FWHM), and area under the curve (AUC) (Fig 5A). Because it is essential to measure similarity in the duration of the contrast concentration until the next wash-out after reaching that maximum, the FWHM was utilized for this purpose. The maximum contrast concentration was defined as the value at which the contrast concentration was the highest. The time-to-peak usually represents the degree of contrast concentration at the selected ROI, and is considered illustrative of the temporal changes in arterial vasculature [11]. Herein, the time-to-peak was defined as the time at which X-ray attenuation reached its maximum level in the angiographic series. The reference time (t = 0) indicates the first instance at which a mask (defined to match the initial skull image) was visible. The first frame was considered at the initial viewing window when there was a difference in contrast. The last frame of the run was the end of the viewing window. The time-to-peak was then recorded as the maximum value that appeared on the time axis. Subsequently, the FWHM and AUC were measured to compare the widths of the curves. The FWHM was defined as the time at which the contrast concentration reached half the maximum value, and the AUC yielded the area under the CCTC graph. We repeated the experiment 10 times in the same ROI (terminal ICA on the frontal plane) of A, B, and C types to obtain median values for the four parameters of the contrast concentration–time curve. These four parameters and overall graph patterns were compared between phantom circuits with and without the plastic cistern and the sponge pad.

Lastly, we compared the pure time delay since this is functionally very important to ventilatory control applications and is essential for achieving a safe and reliable result in dynamic tests for the experimental system (Fig 5A) [13,14].

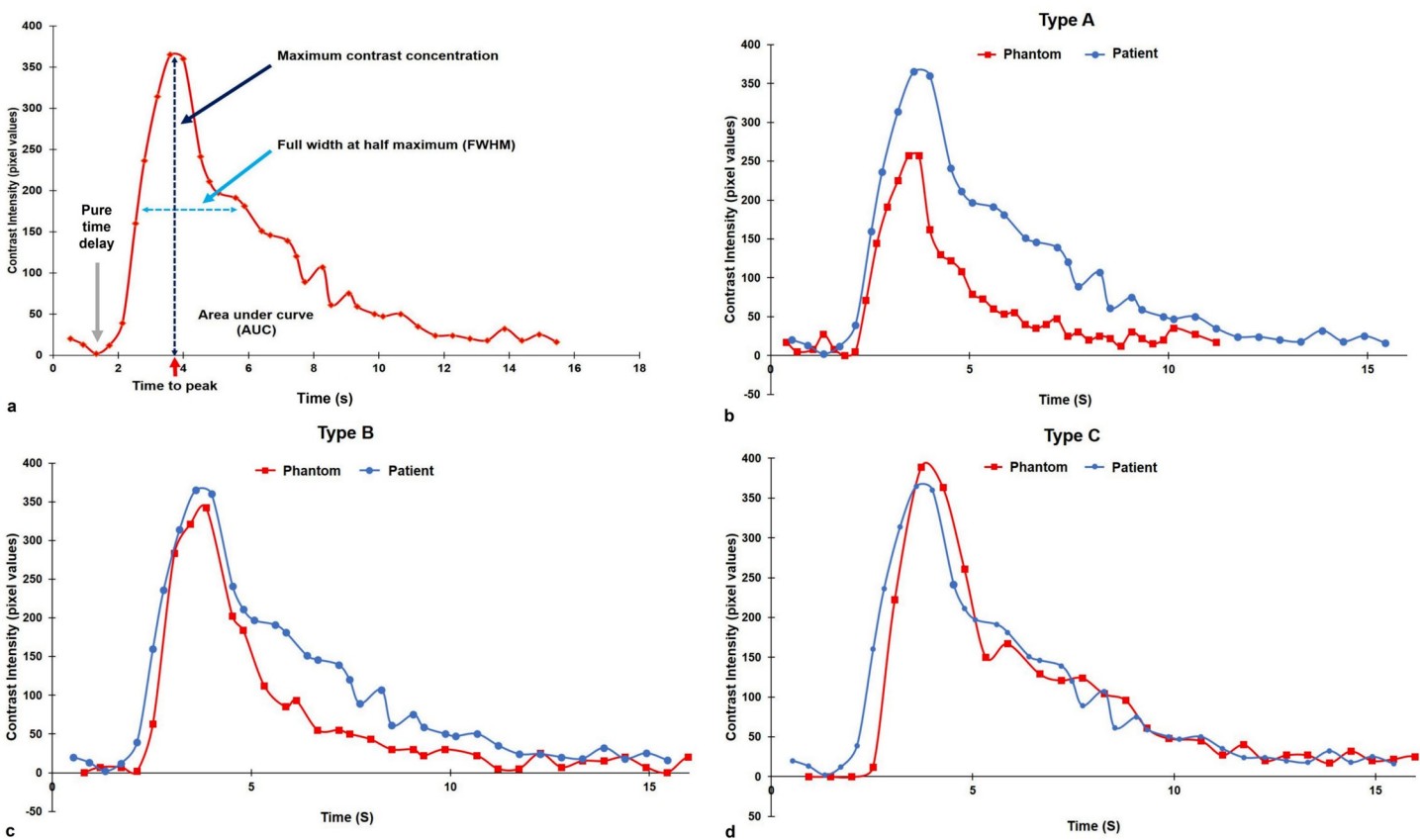

**Fig 5.** Graph descriptions of contrast concentration-time curves (CCTCs) (a) and comparisons between the original control group (blue lines), and three types of cerebrovascular phantoms (red lines) (b–d). (a) The CCTC comprises four parameters (maximum contrast concentration, time-to-peak, full-width-at-half-maximum [FWHM], and area under curve [AUC]). The time-to-peak represents the time at the maximum contrast concentration of the selected ROI. The FWHM and AUC were measured to compare the widths of curves. (b) Pairwise comparisons between the original control group data and (b) the CASP phantom (type A), (c) phantom with plastic cistern only (type C), and (d) plastic cistern with a sponge pad (type C).

## Statistical analysis

All statistical analyses were performed with the use of the software SPSS (version 24.0, IBM). The Wilcoxon signed-rank test (nonparametric method) was utilized to compare the four parameters between each type of cerebrovascular phantom model and the control group data. All P values were two-sided; a P value < .05 was considered statistically significant.

## Results

All four parameters representing the changes in contrast concentration exhibited serial increases in their values in the cases of types A, B, and C (Table 1). We compared differences in each value among the three groups. For the maximum contrast concentration, the median values for types A, B, and C were 271.5 (interquartile range (IQR); 257–276), 364.5 (IQR: 351–377), and 390.0 (IQR: 383–411), respectively. Compared with the control group's maximum contrast concentration, the closest match occurred in the case of type B (P = .168). The time-to-peak value of the control group was 3.60 s; this was the closest to type A (3.74 s, IQR; 3.60–3.87), and there was a significant difference between them (P < .001). In the case of FWHM, the closest value was 2.34 s (IQR: 2.12–2.42) for type C, but there were numerical and statistical differences compared with control group data (P < .001). In the case of the AUC, the median

**Table 1. Four parameters for temporal changes in contrast opacity in the phantom types A, B, and C.**

| | Control group | Type A[†] (n = 10) | P value[*] | Type B[‡] (n = 10) | P value[*] | Type C[§] (n = 10) | P value[*] |
|---|---|---|---|---|---|---|---|
| Maximum contrast concentration | | | | | | | |
| Median (interquartile range [IQR]) | 365.00 | 271.50 (257.00–276.00) | < .001 | 364.50 (351.00–377.00) | 0.168 | 390.00 (383.00–411.00) | < .001 |
| (Min–Max) | | (241.00–298.00) | | (339.00–385.00) | | (300.00–431.00) | |
| Time-to-peak | | | | | | | |
| Median (IQR) | 3.60 | 3.54 (3.47–3.60) | < .001 | 3.74 (3.60–3.87) | < .001 | 3.94 (3.73–4.27) | < .001 |
| (Min–Max) | | (3.33–3.73) | | (3.47–4.00) | | (3.73–4.27) | |
| Full-width-at-half maximum | | | | | | | |
| Median (IQR) | 3.22 | 1.71 (1.68–1.72) | < .001 | 1.87 (1.77–1.96) | < .001 | 2.34 (2.12–2.42) | < .001 |
| (Min–Max) | | (1.61–1.82) | | (1.70–2.10) | | (1.98–4.15) | |
| Area under curve | | | | | | | |
| Median (IQR) | 1519.55 | 657.23 (631.84–720.94) | < .001 | 993.58 (909.79–1062.31) | < .001 | 1572.83 (1492.73–1860.27) | < .001 |
| (Min–Max) | | (592.36–752.22) | | (809.90–1189.80) | | (1313.38–1974.29) | |

[*] P value in comparison with control group values according to Wilcoxon signed-rank test.

[†] Type A: Only with the CASP.

[‡] Type B: CASP with the plastic cistern.

[§] Type C: CASP with the plastic cistern filled with a sponge pad.

value of type C (1572.83, IQR; 1492.73–1860.27) was similar to that of the control group (1519.55), but there was a significant difference between them (P < .001). Despite this statistical result, type C was the most similar one with the FWHM and AUC values closer to those of the control group.

The overall graph patterns of CCTCs of the different groups are shown in Fig 5. The three graphs generated from the median values of contrast concentration measurements determined the shapes of the CCTCs. These plots were compared with the graph of the control group (Fig 5B, 5C and 5D). The DSA pattern of the fluid circuit CASP could be adjusted to mimic that of the control group by adding a sponge pad to the plastic cistern (type C). We further found that the CCTC of type C became broader, showing the widest FWHM and AUC, and higher peak contrast concentration compared with those obtained from types A, and B. As a result, type C showed that the slope of the CCTC graph before and after maximum intensity concentration point was most similar to that of the control group.

The pure time delay was as follows: the control group, 1.33 s; type A, 1.87 s; type B, 2.13 s; type C, 2.00 s; Compared with the control group, the closest match occurred in the case of type A, but the irregular graph pattern was represented. Type B was a more stable graph pattern than that of type A but had the largest difference from the value of the control group among the three types. The pure time delay of type C showed the most constant and stable value among other types (Fig 5). In addition, calculating the difference between the time to peak and the pure time delay, the value showed 2.27 s (control group), 1.67 s (type A), 1.61 s (type B), and 1.94 s (type C), respectively. As a result, type C was confirmed to be closest to the control group.

## Discussion

In this comparative analysis of the fluid circulation CASP and cerebral DSA, we hypothesized that a capacitor formed of a hollow cistern and a sponge pad body could receive blood from the cerebral arterial system on cerebral angiography. As a result, an acrylic hollow cistern filled with a sponge pad was introduced to replicate real contrast filling dynamics. In our study, the

response of the sponge-containing plastic cistern was evaluated by representing the real peripheral resistance system of the brain with a fluid circulation CASP, in conjunction with the Syngo iFlow application. The Syngo iFlow application demonstrated time-dependent changes in the remaining contrast in the large cerebral artery [15–17]. Specifically, the CCTC pattern became broader, and the y-axis values of curves were closer to the control group's graph when each plastic cistern (type B) and a plastic cistern with a sponge pad (type C) were applied than when only CASP was used (type A) (Fig 5). Between the two phantom types (types B vs. C), the DSA pattern of the CASP fluid circuit of type C was most similar to that of the control group because of the addition of a sponge pad to the plastic cistern. Consequently, we confirmed that the resistance generated toward the distal part of the cerebral blood vessel could be reproduced with the artificial capacitor. Quantitatively, four parameters were compared between different types of cerebrovascular phantoms. These parameters could be used to study time-related changes in contrast concentration in the cerebral artery. In the case of maximum contrast concentration and the time-to-peak, type C was not the most similar to that of the control group, but it was marginal from the other two types (A, B). Conversely, type C showed significantly similar values to the control group in the FWHM and the AUC, unlike the other two (type A, and B). Furthermore, the most notable results in our study were the changes of the overall graph patterns. Particularly, from the time-to-peak to the 15 s timepoint, the CCTC of type C yielded the most similar patent to that of the control group data. Along with the facts proven above, the pure time delay of type C was not irregular and showed the most constant and stable pattern, so it was possible to reproduce the blood flow of the cerebral artery most similarly compared to other types. As a result, a plastic cistern filled with a sponge pad could reproduce a lower resistance, replicating real contrast filling dynamics of the cerebral artery.

The cerebral circulation time, which is thought to be closely related to CCTC, can be obtained by measuring the contrast concentration on cerebral angiography. Each cerebral disease is associated with various patterns of angiographic cerebral circulation time. Based on extensive investigations of cerebral arteriography for diseases, such as atherosclerosis and thrombotic cerebral vascular disease, we observed a considerable delay in contrast opacification of small cerebral arteries [18]. A prior study on an occlusive ICA lesion suggested that angiographic cerebral circulation time and cerebral vasoreactivity are well-correlated [19]. Furthermore, a recent study on multiple sclerosis has demonstrated a significant difference in cerebral circulation time between multiple patients with sclerosis (mean = 4.9 s, standard deviation [SD] = 1.27 s) and normal control subjects (mean = 2.8 s, SD = 0.51 s). This suggests that an increased cerebral circulation time in multiple patients with sclerosis corresponds to microvascular dysfunction [20]. Another study has revealed differences in cerebral circulation time between patients with hemorrhagic and non-hemorrhagic moyamoya disease [21]. Cerebral circulation time of the contrast may vary depending on the degree of cerebral perfusion in the brain parenchyma. Additionally, recent studies on dementia have investigated cerebral perfusion in the brain parenchyma [22–24].

Through our study, various cerebrovascular *in vitro* phantoms that can aid in the study of each disease were available by adjusting the transition time of the fluid circulation CASP to reproduce the characteristic cerebral circulation time of a given disease group. When creating an *in vitro* cerebrovascular platform, the key point pertains to the resistance level that should be assigned to the circuit. This fluid circulation of CASP attached to an artificial device, such as the plastic cistern and a sponge pad, is expected to emulate the patient's brain perfusion pattern because the transit time of the intraarterial vasculature can be controlled by adjusting the resistance of the phantom circuit. Future studies are needed to quantitatively express and control resistance in phantom flow circuits.

This study has several limitations. Firstly, the iFlow reference data of the control group were obtained from only one DSA run, and may not be completely accurate. This situation was inevitable considering the ethics of human experimentation, which reflects the importance of the phantom experiment in this study. Although the comparison between phantoms and the single patient dataset was phenomenologically limiting, this was not proven to be a major problem in the application, as the DSA images yielded similar patterns. Secondly, although several studies had investigated the cerebral circulation time of patients with specific diseases (e.g., shunt and steno-occlusion) compared with that of normal patients with color-coded quantitative DSA [11,15], most of the published papers did not directly yield normal graph patterns with the iFlow. Thus, selecting specific patient data as the control group data was considerably tricky. Nevertheless, because the selection of data from the specific patient studied herein was based on careful evaluations and consensus of two observers with extensive clinical experience, the research results are thought to be reliable. Finally, we could not perform detailed examinations based on the control of the amount of a sponge pad. Nevertheless, this study was considered valuable, as it constitutes the first pilot study showing the possibility of emulating contrast filling dynamics with a sponge pad in angiographic studies. It is believed that better realistic graphic curves could be obtained if changes in the amount of sponge and pore size are evaluated in future experiments.

Despite of these limitations in the study design, few experiments have examined the similarities of hemodynamic curves from arterial-pressure-monitoring data obtained from the human body. Ultimately, we were able to verify the feasibility of a silicone vascular phantom replicating real arterial contrast filling dynamics on cerebral angiography. Overall, we were able to develop an *in vitro* phantom that can model the resistance which develops toward the distal parts of blood vessels, which has not been reproduced by several previously developed silicone phantoms.

## Conclusions

Patient CCTC could be simulated in the phantom model by adjustment of the resistance of the phantom flow system using an acrylic hollow cistern filled with a sponge pad. In conclusion, with the hand-made CASPs, we could replicate the contrast filling dynamics of clinical cerebral angiography by applying a small cistern filling with a sponge pad at the venous side of the phantom. In the future, further modifications to the mechanical details of the proposed approach for variable cerebral hemodynamic conditions will be necessary.

## Acknowledgments

We would like to express our gratitude to Byung Joon Lee for his technical help in developing the cerebral arterial silicone phantom), Jaeyoung Kwon for general experimental support, and Ji Sung Lee for statistical analysis application support.

## Author Contributions

**Conceptualization:** Deok Hee Lee.

**Formal analysis:** Ki Baek Lee, Jae Jon Sheen.

**Funding acquisition:** Deok Hee Lee.

**Investigation:** Ki Baek Lee, Jae Jon Sheen, Jong-Tae Yoon, Mi Hyeon Kim, Jun Young Maeng, Sun Moon Hwang, Joon Ho Choi.

**Methodology:** Jae Jon Sheen, Deok Hee Lee.

**Supervision:** Deok Hee Lee.

**Validation:** Ki Baek Lee, Jun Young Maeng, Sun Moon Hwang, Joon Ho Choi.

**Visualization:** Ki Baek Lee, Jong-Tae Yoon, Mi Hyeon Kim.

**Writing – original draft:** Ki Baek Lee, Jae Jon Sheen.

**Writing – review & editing:** Ki Baek Lee, Deok Hee Lee.

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
