## [Decision Letter · Decision Letter 0]

13 Sep 2021

PONE-D-21-24998Feasibility of a silicone vascular phantom replicating real arterial contrast filling dynamics on cerebral angiographyPLOS ONE

Dear Dr. Lee,

Thank you for submitting your manuscript to PLOS ONE. After careful consideration, we feel that it has merit but does not fully meet PLOS ONE’s publication criteria as it currently stands. Therefore, we invite you to submit a revised version of the manuscript that addresses the points raised during the review process.

Both the Reviewers and the Editor were satisfied to see the Authors` selection of an important and practically treated simulation problem. Please try to insert some reasoning as per the selection of the sponge for venous resistor model, if possible reasoned by some physiology or neuroscience books and papers, to have the readership a good understanding of the difficulties in modeling but also to see how the seemingly specific information in your treatise can be generalised for other brain vasculature-related tasks.Please be sure to answer to comments of Reviewer 2 especially about the silicone material.Please, then submit the following as revision:A rebuttal letter that responds to each point raised by the academic editor and reviewer(s). You should upload this letter as a separate file labeled 'Response to Reviewers'.A marked-up copy of your manuscript that highlights changes made to the original version. You should upload this as a separate file labeled 'Revised Manuscript with Track Changes'.An unmarked version of your revised paper without tracked changes. You should upload this as a separate file labeled 'Manuscript'.

We look forward to receiving your revised manuscript.

Kind regards,

Domokos Máthé DVM, PhD

Academic Editor

PLOS ONE

Journal Requirements:

 [D.H.L. This work was supported by the Advanced Technology Center Plus (ATC+) Program (Grant numbers: 20009512) funded by the Ministry of Trade, Industry & Energy (MOTIE, Korea), and by the Technology Development Program (Grant numbers: S2836138) funded by the Ministry of SMEs and Startups (MSS, Korea). No.]

[No]. 

Additional Editor Comments:

I am pleased to see this work, and I invite the Authors to work together with the Reviewers and myself to produce a truly ciseled and interesting paper at the end of this publication process.

Reviewers' comments:

Reviewer's Responses to Questions

**Comments to the Author**

1. Is the manuscript technically sound, and do the data support the conclusions?

Reviewer #1: Yes

Reviewer #2: Partly

2. Has the statistical analysis been performed appropriately and rigorously? 

Reviewer #1: N/A

Reviewer #2: Yes

3. Have the authors made all data underlying the findings in their manuscript fully available?

Reviewer #1: Yes

Reviewer #2: No

4. Is the manuscript presented in an intelligible fashion and written in standard English?

Reviewer #1: Yes

Reviewer #2: Yes

5. Review Comments to the Author

Reviewer #1: This is an excellent pilot study about contrast-filling dynamics in cerebral angiography. I recommend to include in your title: Pilot study.

How the authors come up with the idea of a sponge to be used as vascular resistance? Are there any other experiments where they used the same material? Can the authors mention in the paper?

Reviewer #2: Initially, I would like to congratulate the authors for their interest in advancing in this unexplored and still extremely relevant area.

Understanding vascular hemodynamics for in vitro experimentation is an arduous and complex task. Thanks again to the authors for the initiative.

Comments:

1. Introduction:

- Page 3, lines 44-46: change the arrangement of information. Start with in vitro studies and then comment on preclinical work.

2. Materials and Methods:

- Page 3, line 67: specify how many patients were angiographically evaluated and subsequently excluded;

- Page 4, line 72: specify how the chosen patient was evaluated. Electronic medical record review? Physical examination?;

- Page 4, line 75: make it clear that only one patient was used in the study;

- Page 4, line 77: emphasize to the reader that the description of 3D printing will be further detailed later in the Materials and Methods section;

- Page 5, line 98: when it comes to imaging terms, use descriptors such as "axial, sagittal and coronal planes". In the specific case, coronal plane;

- Page 5, line 116: what type or mixture was obtained for the liquid silicone used? Is this product commercially available or was it manufactured by the authors? If commercially available, add brand and specifications. If manufactured, provide details of the mixture and components;

- Page 7, line 147: Has the 40:60 glycerol/water mixture been validated by previous studies? Clarify why this specific dilution was used;

- Page 8, line 172: Has FWHM been used in other studies? Clarify for the reader how the authors arrived at this parameter and the importance of its use.

3. Results and discussion:

- For better understanding and standardization, consider dividing the sections into Results / Discussion;

- Virtually all data were statistically different from the control group. In this sense, it should be noted that, despite this statistical result, group C was the one with AUC values closer to A. Draw the reader's attention to the favorable data in group C.

4. Minor technical comments:

- Eliminate the use of informal expressions. Example: abstract "to figure out", "simply by"; line 74 "we figured out"; conclusion "there is room for".

Once again, I would like to congratulate the authors for making progress in this very important field.

We await revisions. Thank You.

6. PLOS authors have the option to publish the peer review history of their article (what does this mean?). If published, this will include your full peer review and any attached files.

Reviewer #1: No

Reviewer #2: **Yes: **Marcio dos Santos Meira

---

## [Author Response · Author response to Decision Letter 0]

7 Oct 2021

First of all, we would like to thank the reviewers for a thorough reading of the manuscript and helpful comments and suggestions. We have revised our manuscript based on the comments made by the reviewer, and we believe the manuscript is now significantly improved. 

Here we are gladly submitting a revised version of the manuscript (Manuscript ID: PONE-D-21-24998) entitled, “Feasibility of a silicone vascular phantom replicating real arterial contrast filling dynamics on cerebral angiography: an in-vitro pilot study” to be considered for publication in PLoS One. We hope that our revised version will be considered for publication in your journal. 

Thank you very much for your time and efforts that go into the publication of this paper

---

## [Decision Letter · Decision Letter 1]

25 Mar 2022

PONE-D-21-24998R1Feasibility of a silicone vascular phantom replicating real arterial contrast filling dynamics on cerebral angiography: an in-vitro pilot studyPLOS ONE

Dear Dr. Lee,

Thank you for submitting your manuscript to PLOS ONE. After careful consideration, we feel that it has merit but does not fully meet PLOS ONE’s publication criteria as it currently stands. Therefore, we invite you to submit a revised version of the manuscript that addresses the points raised during the review process.

Reviewers recognize the interest of this paper but made some comments on scientific points and quality of the edition for improving the paper. Please respond point by point to the comments.

We look forward to receiving your revised manuscript.

Kind regards,

Alain-Pierre Gadeau, Ph.D

Academic Editor

PLOS ONE

Reviewers' comments:

Reviewer's Responses to Questions

**Comments to the Author**

1. If the authors have adequately addressed your comments raised in a previous round of review and you feel that this manuscript is now acceptable for publication, you may indicate that here to bypass the “Comments to the Author” section, enter your conflict of interest statement in the “Confidential to Editor” section, and submit your "Accept" recommendation.

Reviewer #3: All comments have been addressed

Reviewer #4: (No Response)

2. Is the manuscript technically sound, and do the data support the conclusions?

Reviewer #3: Yes

Reviewer #4: Partly

3. Has the statistical analysis been performed appropriately and rigorously? 

Reviewer #3: Yes

Reviewer #4: Yes

4. Have the authors made all data underlying the findings in their manuscript fully available?

Reviewer #3: Yes

Reviewer #4: No

5. Is the manuscript presented in an intelligible fashion and written in standard English?

Reviewer #3: Yes

Reviewer #4: No

6. Review Comments to the Author

Reviewer #3: The response analysis considered time to peak and temporal shape of the response, but did not include pure time delay which is critical to control of breathing studies. For this purpose the response should be analyzed with 3 parameters: pure time delay and 2 time constants. See Lange J Appl Physiol 1966:21:1281-1291.

Reviewer #4: Dear Authors,

First, I would like to congratulate the authors for this publication and the development of a setup including flow phantoms which are pertinent to develop clinical protocol before the application on patients.

However, there are a number of issues to be addressed:

1. The manuscript needs a comprehensive rewrite/ review by a native English speaker.

2. In the introduction section the problematic and the importance of phantoms development is not clearly defined, moreover the introduction is poorly referenced.

3. The organization of the section material & methods is quite confusing. For example, the sentence (p4, l85) is not clear as we are already in the material & methods section.

4. A flow diagram will be interesting to understand the patient selection.

5. P5, l99, reference number does not follow same model as the others numbers.

6. P7, l153. Would it be possible to perform analysis with others pulse rates? Beats per minute are physiologically at 60, however relatively to the disease it can increase and it would be interesting to investigate the performance with different pulse rate.

7. In results section, please add 95 % for confidence interval or interquartile range.

7. PLOS authors have the option to publish the peer review history of their article (what does this mean?). If published, this will include your full peer review and any attached files.

Reviewer #3: No

Reviewer #4: No

---

## [Author Response · Author response to Decision Letter 1]

9 May 2022

We very much appreciate each reviewer’s insightful suggestions directed to enhancing the quality of our work. In the response to reviewers's page, we provided all the items revised according to the reviewers’ comments.

---

## [Decision Letter · Decision Letter 2]

26 May 2022

PONE-D-21-24998R2Feasibility of a silicone vascular phantom replicating real arterial contrast filling dynamics on cerebral angiography: an in-vitro pilot studyPLOS ONE

Dear Dr. Lee,

Thank you for submitting your manuscript to PLOS ONE. After careful consideration, we feel that it has merit but does not fully meet PLOS ONE’s publication criteria as it currently stands. Therefore, we invite you to submit a revised version of the manuscript that addresses the points raised during the review process.

You are on a good way for publishing this study: please add a separation of pure time delay to your analysis as suggested by one reviewer! 

We look forward to receiving your revised manuscript.

Kind regards,

Stephan Meckel, MD, PhD

Academic Editor

PLOS ONE

Journal Requirements:

Reviewers' comments:

Reviewer's Responses to Questions

**Comments to the Author**

1. If the authors have adequately addressed your comments raised in a previous round of review and you feel that this manuscript is now acceptable for publication, you may indicate that here to bypass the “Comments to the Author” section, enter your conflict of interest statement in the “Confidential to Editor” section, and submit your "Accept" recommendation.

Reviewer #3: All comments have been addressed

Reviewer #4: All comments have been addressed

2. Is the manuscript technically sound, and do the data support the conclusions?

Reviewer #3: Yes

Reviewer #4: Yes

3. Has the statistical analysis been performed appropriately and rigorously? 

Reviewer #3: Yes

Reviewer #4: Yes

4. Have the authors made all data underlying the findings in their manuscript fully available?

Reviewer #3: Yes

Reviewer #4: Yes

5. Is the manuscript presented in an intelligible fashion and written in standard English?

Reviewer #3: Yes

Reviewer #4: Yes

6. Review Comments to the Author

Reviewer #3: Separating pure time delay has not been done yet would enhance this paper. Pure time delay is functionally very important to ventilatory control applications.

Reviewer #4: (No Response)

7. PLOS authors have the option to publish the peer review history of their article (what does this mean?). If published, this will include your full peer review and any attached files.

Reviewer #3: No

Reviewer #4: No

---

## [Author Response · Author response to Decision Letter 2]

9 Jun 2022

Reviewer #3: Separating pure time delay has not been done yet would enhance this paper. Pure time delay is functionally very important to ventilatory control applications.

Response: In fact, we recognize that the value of pure time delay you mentioned are functionally very important for ventilation control applications. So, it seems that pure time delay analysis is essential if it is the chest part, or the area closely related to respiration. However, the region of interest in this study is the cerebral artery in the brain, which is relatively insensitive to respiration. In addition, it is judged that the effect on this is small because the image acquisition proceeded with the patient holding the breath. 

Also, this study tried to reproduce the actual flow of the cerebral artery with a silicone model, and in particular, it was reproduced to create a shape that gradually decreases after reaching the time-to-peak similarly to the human head. In other words, the most important consideration is to reproduce the time-to-peak value and the wash-out degree similarly (refer to the figure below - Type C is the most ideal). Therefore, pure time delay analysis was not performed separately, and this content has been presented in the discussion section. At this time, we added two reference papers on pure time delay analysis.

---

## [Decision Letter · Decision Letter 3]

2 Sep 2022

PONE-D-21-24998R3Feasibility of a silicone vascular phantom replicating real arterial contrast filling dynamics on cerebral angiography: an in-vitro pilot studyPLOS ONE

Dear Dr. Lee,

Thank you for submitting your manuscript to PLOS ONE. After careful consideration, we feel that it has merit but does not fully meet PLOS ONE’s publication criteria as it currently stands. Therefore, we invite you to submit a revised version of the manuscript that addresses the points raised during the review process.

The comments of reviewer 3 were not answered correctly. We cannot publish your manuscript without changes being made correctly according to his comments! 

We look forward to receiving your revised manuscript.

Kind regards,

Stephan Meckel, MD, PhD

Academic Editor

PLOS ONE

Journal Requirements:

Reviewers' comments:

Reviewer's Responses to Questions

**Comments to the Author**

1. If the authors have adequately addressed your comments raised in a previous round of review and you feel that this manuscript is now acceptable for publication, you may indicate that here to bypass the “Comments to the Author” section, enter your conflict of interest statement in the “Confidential to Editor” section, and submit your "Accept" recommendation.

Reviewer #3: (No Response)

2. Is the manuscript technically sound, and do the data support the conclusions?

Reviewer #3: Yes

3. Has the statistical analysis been performed appropriately and rigorously? 

Reviewer #3: Yes

4. Have the authors made all data underlying the findings in their manuscript fully available?

Reviewer #3: Yes

5. Is the manuscript presented in an intelligible fashion and written in standard English?

Reviewer #3: No

6. Review Comments to the Author

Reviewer #3: The comments added to the revision are not correct. The region of interest being "relatively insensitive to respiration" and "patient holding the breath" have nothing to do with circulatory delay which is due to blood flow.. From the response figure of concentration versus time the pure time delay is the time from 0 time to when concentration is just detectable above baseline. This can be measured along with time to peak as indicated in the Lange reference which has been added in the revision.

7. PLOS authors have the option to publish the peer review history of their article (what does this mean?). If published, this will include your full peer review and any attached files.

Reviewer #3: No

---

## [Author Response · Author response to Decision Letter 3]

15 Oct 2022

Reviewer #3: Separating pure time delay has not been done yet would enhance this paper. Pure time delay is functionally very important to ventilatory control applications.

Response: Thank you for your great comment, we added the pure time delay values of the control and experimental groups. In addition, two reference papers on pure time delay have been added to the manuscript. As a result, we revealed the pure time delay of type C showed the most constant and stable pattern, so it was possible to reproduce the blood flow of the cerebral artery most similarly compared to other types.

---

## [Decision Letter · Decision Letter 4]

29 Nov 2022

PONE-D-21-24998R4Feasibility of a silicone vascular phantom replicating real arterial contrast filling dynamics on cerebral angiography: an in-vitro pilot studyPLOS ONE

Dear Dr. Lee,

Thank you for submitting your manuscript to PLOS ONE. After careful consideration, we feel that it has merit but does not fully meet PLOS ONE’s publication criteria as it currently stands. Therefore, we invite you to submit a revised version of the manuscript that addresses the points raised during the review process.

ssee my author comment below!==============================

We look forward to receiving your revised manuscript.

Kind regards,

Stephan Meckel, MD, PhD

Academic Editor

PLOS ONE

Journal Requirements:

Additional Editor Comments (if provided):

Please address the additional comment of reviewer 3 regarding the time delay and modify or delete the respective sentence of your discussion!

Reviewers' comments:

Reviewer's Responses to Questions

**Comments to the Author**

1. If the authors have adequately addressed your comments raised in a previous round of review and you feel that this manuscript is now acceptable for publication, you may indicate that here to bypass the “Comments to the Author” section, enter your conflict of interest statement in the “Confidential to Editor” section, and submit your "Accept" recommendation.

Reviewer #3: All comments have been addressed

2. Is the manuscript technically sound, and do the data support the conclusions?

Reviewer #3: Yes

3. Has the statistical analysis been performed appropriately and rigorously? 

Reviewer #3: Yes

4. Have the authors made all data underlying the findings in their manuscript fully available?

Reviewer #3: Yes

5. Is the manuscript presented in an intelligible fashion and written in standard English?

Reviewer #3: (No Response)

6. Review Comments to the Author

Reviewer #3: On page 15 The sentence beginning with: " However, we did not analyze the time delay..." is misleading and should be changed. The time delay as shown by others is not due to respiration but blood flow. The time delay as now measured was about half the time to peak for group C- 2 sec compared to 4.

7. PLOS authors have the option to publish the peer review history of their article (what does this mean?). If published, this will include your full peer review and any attached files.

Reviewer #3: No

---

## [Author Response · Author response to Decision Letter 4]

30 Nov 2022

Reviewer #3: On page 15 The sentence beginning with: " However, we did not analyze the time delay..." is misleading and should be changed. The time delay as shown by others is not due to respiration but blood flow. The time delay as now measured was about half the time to peak for group C- 2 sec compared to 4.

Response: Thank you for your great comment, we deleted the sentence beginning with: "However, we did not analyze the time delay...". In fact, we should have deleted the misleading sentence in the previous revision manuscript, but accidentally failed to make the change. Thanks again for letting us know. In addition, we tried to increase the reliability of the results by adding a sentence showing the difference between the time to peak and the pure time delay by the group in the result part (i.e., 2.27 s; the control group, 1.67 s; type A, 1.61 s; type B, and 1.94 s; type C).

---

## [Decision Letter · Decision Letter 5]

28 Dec 2022

Feasibility of a silicone vascular phantom replicating real arterial contrast filling dynamics on cerebral angiography: an in-vitro pilot study

PONE-D-21-24998R5

Dear Dr. Lee,

We’re pleased to inform you that your manuscript has been judged scientifically suitable for publication and will be formally accepted for publication once it meets all outstanding technical requirements.

Kind regards,

Stephan Meckel, MD, PhD

Academic Editor

PLOS ONE

Additional Editor Comments (optional):

Reviewers' comments:

Reviewer's Responses to Questions

**Comments to the Author**

1. If the authors have adequately addressed your comments raised in a previous round of review and you feel that this manuscript is now acceptable for publication, you may indicate that here to bypass the “Comments to the Author” section, enter your conflict of interest statement in the “Confidential to Editor” section, and submit your "Accept" recommendation.

Reviewer #3: All comments have been addressed

2. Is the manuscript technically sound, and do the data support the conclusions?

Reviewer #3: Yes

3. Has the statistical analysis been performed appropriately and rigorously? 

Reviewer #3: Yes

4. Have the authors made all data underlying the findings in their manuscript fully available?

Reviewer #3: Yes

5. Is the manuscript presented in an intelligible fashion and written in standard English?

Reviewer #3: Yes

6. Review Comments to the Author

Reviewer #3: All required questions have been answered and all responses meet formatting specifications. It is now suitable for publication.

7. PLOS authors have the option to publish the peer review history of their article (what does this mean?). If published, this will include your full peer review and any attached files.

Reviewer #3: No

---

## [Editor Report · Acceptance letter]

5 Jan 2023

PONE-D-21-24998R5 

Feasibility of a silicone vascular phantom replicating real arterial contrast filling dynamics on cerebral angiography: an in-vitro pilot study 

Dear Dr. Lee:

I'm pleased to inform you that your manuscript has been deemed suitable for publication in PLOS ONE. Congratulations! Your manuscript is now with our production department. 

Kind regards, 

on behalf of

Prof. Dr. Stephan Meckel 

Academic Editor

PLOS ONE